# Bacterial profile and antimicrobial susceptibility patterns in cancer patients

**Minichil Worku⊙, Gizeaddis Belay⊙, Abiye Tigabu⊙* **

Department of Medical Microbiology, School of Biomedical and Laboratory Sciences, University of Gondar, Gondar, Ethiopia

⊙ These authors contributed equally to this work.
* abty12@gmail.com

## Abstract

### Background

Bloodstream infections have been the leading complications in cancer patients because they are at high risk for antibiotic-resistant bacterial infections. There is increasing evidence from different parts of the world of the high prevalence of antimicrobial-resistant bacterial strains in cancer patients. The burden of the infection is high in developing countries, especially in Ethiopia. Data on bacterial profile and antimicrobial susceptibility patterns among cancer patients in Ethiopia is limited. Thus, this study aimed to determine the predominant bacterial species causing bacteremia and their antibiotic resistance pattern among cancer patients at University of Gondar comprehensive specialized hospital.

### Methods

A hospital-based, cross-sectional study was conducted on 200 study participants from March to July 2021. All cancer patients who developed a fever at the time of hospital visit were included in this study, and their socio-demographic and clinical data were collected using a structured questionnaire. Blood samples (10 mL for adults and 4 mL for children) were collected from each patient, and the collected blood samples were transferred into sterile tryptic soy broth, then incubated at 37°C for 7 days. Tryptic soy broth which showed signs of growth were Gram-stained and sub-cultured on blood agar, chocolate agar, Mac-Conkey agar, and mannitol salt agar. The inoculated plates were then aerobically incubated at 37°C for 18–24 hours and the isolates obtained were identified using standard microbiological methods. Antimicrobial susceptibility tests were done using a modified Kirby-Bauer disk diffusion technique following CLSI 2021 guidelines. Data were entered using EPI data version 4.6 and analyzed with SPSS version 20.

### Results

In this study, out of 200 cancer patients included and 67.5% (135/200) of them were males. The majorities of study participants, 56% (113/200) of cancer patients were pediatrics and 26.5% (53/200) of them belong under five years of age. Out of 200 patient samples that had undergone culture, 27% (54/200) samples had bacterial growth. Gram-positive bacterial

**Data Availability Statement:** All relevant data are within the manuscript and its supporting information files.

**Funding:** The author(s) received no specific funding for this work.

**Competing interests:** The authors have declared that no competing interests exist.

**Abbreviations:** ATCC, American type culture collection; BSI, Blood stream infection; CLSI, Clinical Laboratory Standards Institute; CoNS, Coagulase negative staphylococci; GPB, Gram positive bacteria; GNB, Gram negative bacteria; MHA, Muller Hinton Agar; MDR, Multidrug-Resistant; UoGCSH, University of Gondar Comprehensive Specialized Hospital.

isolates were predominant, 61.1%, and *S. aureus* was the predominant Gram-positive isolate, (51.5.6%), followed by coagulase-negative staphylococci (48.5%). Moreover, *K. pneumoniae* (47%) and *P. aeruginosa* (29.5%) were the most common Gram-negative bacterial isolates. Among patients who had BSIs, the highest prevalence of BSIs was observed among males (66.7%), and in pediatrics cancer patients (44.2%). Pediatric study participants were more venerable to bloodstream infection (P = 0.000) compared to adult participants. Meropenem (100%), amikacin (100%), piperacillin/tazobactam (72.3%), and ceftazidime (73.5%) were effective against for Gram-negative isolates while cefoxitin (81.2%) and penicillin (70.5%) were effective for Gram-positive isolates. Additionally, most Gram-negative and Gram-positive bacterial isolates were sensitive for gentamycin (75.9%). Multidrug resistance was seen among 17.1% bacterial isolates, and MDR in Gram-negative and Gram-positive bacteria were 83.3% and 16.7%, respectively. Gram-negative bacterial isolates showed a high prevalence of MDR than Gram-positive isolates.

## Conclusions and recommendation

BSI's remains an important health problem in cancer patients, and Gram-positive bacteria were more common as etiologic agents of BSIs in cancer patients. *S. aureus* was the dominant bacteria followed by CoNS, *K. pneumoniae*, and *P. aeruginosa*. Multidrug-resistant isolates found in cancer patients and routine bacterial surveillance and study of their resistance patterns may guide successful antimicrobial therapy and improve the quality of care. Therefore, strict regulation of antibiotic stewardship and infection control programs should be considered in the study area.

## Background

Oncology patients on chemotherapy are at particular risk for bacterial, and fungal infections and bloodstream infection remains the leading cause of morbidity and mortality in patients undergoing treatment for cancer which causes life-threatening complications and a significantly elevated risk of infection-related death [1–3]. More than half of all cancer cases and about 60% of deaths occur in developing countries [4, 5]. Infection-related mortality affects the survival rates of patients who are receiving treatment for cancer [6].

Blood-stream infections in oncology patients become a growing problem in sub-Saharan African countries due to immunodeficiencies, immunosuppressive anticancer chemotherapy, and frequent interaction with health care settings for treatment [7]. Neutropenia, emerging advanced life-support facilities, altered gut flora, disruption of skin, and damage of epithelial surfaces increase the susceptibility of cancer patients to infection [8]. Neutropenia is an important risk factor associated with BSIs in up to 25% of cancer patients, with death rates as high as 24% in high-income countries and 33% in low-middle income countries [9–11].

Moreover, BSI's represents about 15% of all nosocomial infections, and infections emerged in cancer patients cause a disorder of the treatment pattern, hospital stay, and an increase in treatment cost as well as a reduced survival rate in patients [8, 12–15]. Bacterial BSIs and antimicrobial resistance frequently result in treatment failure and prolonged infections in cancer patients [16–18]. The most common bacterial isolates from patients with cancer were *S. aureus*, *A. baumannii*, *K. pneumoniae*, *P. aeruginosa*, and *E. coli* [1–3]. *S. aureus* is one of the major pathogens of humans causes various suppurative diseases, food poisoning, pneumonia,

sepsis, and toxic shock syndrome by producing many virulence factors, including toxins (α-toxin, β-toxin, δ-toxin, P-V Leukocidin, enterotoxin, exfoliative toxin, and toxic shock syndrome toxin), enzymes (coagulase, proteases, lipase, hyaluronidase, staphylokinase, and nuclease), antigens (capsule, protein A, fibrin binding protein, and adhesins), immune-modulatory factors, and iron acquisition factors [19–21].

*K. pneumoniae* is one of the leading causes of neonatal sepsis and the most common cause of bacteremia, pneumonia, liver abscess, meningitis, wound infections, purulent abscesses, urinary tract infection, and endophthalmitis. Capsule, mucoid phenotype, iron acquisition systems, and beta-lactamases production are important determinants for virulence of *K. pneumoniae* [22–24]. Additionally, *P. aeruginosa* is an increasingly prevalent opportunistic pathogen and responsible for severe life-threatening nosocomial infections such as nosocomial pneumonia cases, hospital-acquired urinary tract infections, surgical wound infections, and bloodstream infections. *P. aeruginosa* possesses cell-associated (flagella, pili, lectins, alginate/biofilm, lipopolysaccharide) and extracellular (proteases, hemolysins, cytotoxin, pyocyanin, siderophores, exotoxin A, exoenzyme S, exoenzyme U, motility) virulence factors [25–27].

The World Health Organization (WHO) has identified antimicrobial resistance as one of the greatest threats to human health, and it becomes challenging in immunocompromised cancer patients [28, 29]. One of the main concerns in BSIs is the development of multi-drug resistance which is a significant cause of morbidity and mortality, especially in hospitalized patients [1]. Frequent and irrational use of antibiotics and prolonged hospital stays lead to a high prevalence of antimicrobial resistance in cancer patients, which calls urgent measures to optimize the application of antibiotics in cancer patients to prevent further increase of antibiotic resistance [30, 31]. Antimicrobial susceptibility patterns of bacterial isolates changed in recent years, due to the increased and incorrect use of antibiotics. Treating bacterial infection in cancer patients is clinically challenging due to an increasing level of resistance for most of the antibiotics used for empiric therapy. There is an emergence of multidrug-resistant *K. pneumoniae*, *P. aeruginosa*, *A. baumannii*, and *S. aureus* isolates in clinical settings [6, 32, 33].

Multidrug resistance has been increased throughout the world that is considered a public health threat [34, 35]. Several recent investigations reported the emergence of multidrug-resistant bacterial pathogens from different origins including humans, birds, cattle, and fish that increase the need for routine application of the antimicrobial susceptibility testing to detect the antibiotic of choice as well as the screening of the emerging MDR strains [36–40]. Antibiotic resistance is a growing problem in developing countries and in Ethiopia, the unregulated sale of antimicrobials leads to the emergence and rapid dissemination of resistant bacterial strains [41]. Rapid and reliable detection of pathogens and their antibiotic susceptibility patterns is crucial in the management of septic patients [42]. Therefore, understanding bacterial profile and antibiotic resistance for BSI is imperative to inform clinical practice and stewardship in relation to the appropriate use of antibiotics in cancer patients.

## Materials and methods

### Study area

The study was conducted at University of Gondar comprehensive specialized hospital (UoGCSH), located in Gondar city, Ethiopia, providing surgical, medical, pediatric, gynecologic, obstetric, oncologic, and ophthalmologic services for more than seven million people. It is a multidisciplinary specialized hospital with 700 inpatient beds and consists of an operating room, intensive care units, fistula center, different wards, and outpatient departments. The cancer treatment center of UoGCSH provides services for more than 2,300 cancer patients, and it has outpatient and inpatient departments with 20 beds.

## Study design and period

A hospital-based cross-sectional study was conducted at UoGCSH from March to July 2021. All cancer confirmed patients who visited the UoGCSH cancer treatment center were the study population. Cancer patients who developed a fever at the time of hospital visits were included in the study period. However, study participants who were unable to give socio-demographic information and blood samples at the time of data collection were excluded.

## Sample size determination and sampling techniques

The sample size was determined using a single population proportion formula using the prevalence reported at UoGCSH 19%, precision level 5%, and confidence interval 95%. The calculated sample size was 236, and we have got only 200 cancer patients during the study period, and a total of 200 samples were collected using a convenient sampling technique.

## Bacterial isolation and identification

Blood samples (10 mL for adults and 4 mL for children) were collected from each patient who developed a fever at the time of diagnosis by experienced nurses before any antibiotic use [43]. Then the collected blood samples were transferred into sterile tryptic soy broth (Oxoid Ltd., Basingstoke, UK) and incubated at 37˚C for seven days [44]. Bacteria growth indicators such as turbidity, hemolysis, clot formation checked on a daily bias for up to seven days, and tryptic soy broth (Oxoid Ltd., Basingstoke, UK) that showed signs of growth were Gram-stained and sub-cultured on blood agar (Oxoid Ltd., Basingstoke, UK), chocolate agar (Oxoid Ltd., Basingstoke, UK), MacConkey agar (Oxoid Ltd., Basingstoke, UK), and mannitol salt agar (Oxoid Ltd., Basingstoke, UK). After that, the inoculated culture plates were incubated aerobically at 37˚C for 18–24 hours, and the isolates identified using standard microbiological methods [45].

Gram-negative bacteria were identified using a series of biochemical tests such as indole test, urease test, lysine decarboxylase test, triple sugar iron agar, citrate utilization, and motility tests. On the other hand, Gram-positive bacteria were identified based on Gram reaction, hemolytic pattern, catalase, and coagulase tests [45]. The isolates that were golden yellow colony on mannitol salt agar (Oxoid Ltd., Basingstoke, UK) and blood agar plate (Oxoid Ltd., Basingstoke, UK), Gram-positive cocci in clusters, catalase, and coagulase-positive were confirmed as *S. aureus* while isolates that were white colony blood agar plate (Oxoid Ltd., Basingstoke, UK), Gram-positive cocci, catalase positive and coagulase negative were confirmed as Coagulase-negative staphylococci (CoNS) [46].

## Antimicrobial susceptibility testing

An antimicrobial susceptibility test was carried out for each bacterial isolate using the Kirby–Bauer disc diffusion method. Briefly, three to five selected pure colonies were taken and transferred to a tube containing 5 ml of sterile normal saline and mixed gently to form a homogeneous suspension until the turbidity of suspension becomes adjusted to 0.5 McFarland standards. Then, using sterile cotton-tipped swabs, the bacteria distribute evenly over the entire surface of Mueller-Hinton agar (Oxoid Ltd., Basingstoke, UK). Then, the inoculated plates were left at room temperature for 15 minutes and using sterile forceps a set of antibiotic discs were placed on the inoculated MHA plates [43, 45].

Antimicrobials were selected according to Clinical Laboratory Standard Institute guideline (CLSI 2021) [47] and the antimicrobial classes were Penicillins (piperacillin/tazobactam (100/10μg, Oxoid Ltd., Basingstoke, UK), amoxicillin/clavulanic acid (20/10μg, Oxoid Ltd., Basingstoke, UK), ampicillin (10μg, Oxoid Ltd., Basingstoke, UK), penicillin (10μg, Oxoid Ltd.,

Basingstoke, UK)), Aminoglycosides (gentamicin (10μg, Oxoid Ltd., Basingstoke, UK), amikacin (30μg, Oxoid Ltd., Basingstoke, UK), Tobramycin (10μg, Oxoid Ltd., Basingstoke, UK)), Fluoroquinolones (ciprofloxacin (5μg, Oxoid Ltd., Basingstoke, UK)), Folate pathway inhibitors (trimethoprim-sulphamethoxazole (SXT) (25μg, Oxoid Ltd., Basingstoke, UK)), Glycopeptides (vancomycin (30μg, Oxoid Ltd., Basingstoke, UK), Tetracycline's (tetracycline (30μg, Oxoid Ltd., Basingstoke, UK)), Carbapenems (meropenem (10μg, Oxoid Ltd., Basingstoke, UK)), Cephamycins (cefoxitin (30μg, Oxoid Ltd., Basingstoke, UK)), and Cephalosporins (ceftriaxone (30μg, Oxoid Ltd., Basingstoke, UK), ceftazidime (30μg, Oxoid Ltd., Basingstoke, UK)).

After placing the antibiotic discs, the MHA plates were allowed to stand for another 15 minutes at room temperature to dissolve antibiotics in the media. Then, plates were incubated at 37˚C for 18 to 24 hours. Finally, zones of inhibitions were measured using a ruler and interpreted according to CLSI 2021 guidelines [47]. Cefoxitin susceptibility of *S. aureus* isolates was tested by placing cefoxitin (30μg) antibiotic discs on MHA using the Kirby–Bauer disc diffusion method and then incubated aerobically at 35°C for 24 hours. Finally, *S. aureus* isolates zone of inhibition $\geq$ 22 mm classified as susceptible and $\leq$ 21 as non-susceptible. *S. aureus* isolates resistant to cefoxitin ($\leq$ 21 mm) were confirmed as MRSA [48].

## Data quality control and analysis

Sample collection, transportation, bacteriological cultivation, and biochemical tests were done according to standard microbiological procedures. Five percent (5%) of the prepared culture media were randomly selected and incubated aerobically for 24 hours at 37˚C to cheek the sterility of culture media. In addition, known strains of *S. aureus* (ATCC 25923) and *E. coli* (ATCC 25922) were inoculated to check the performance of the prepared culture media. Inoculation of culture media, colony characterization, and measurement of susceptibility test was checked by an experienced microbiologist. To standardize the density of the inoculum of bacterial suspension, 0.5 McFarland turbidity standard was used. All data were checked for completeness, entered by EPI data version 4.6, and analyzed with SPSS version 20. Binary and multivariate logistic regression was done to determine the association between independent and dependent variables. A p-value $< 0.05$ was considered statistically significant at 95% confidence interval.

## Ethics approval and consent to participate

Ethical clearance was obtained from the ethical review committee of the School of Biomedical and Laboratory Sciences, University of Gondar (Ref. no. SBMLS/3024). Written informed consent from adults and consent from parents/guardians of the minors were obtained. Additionally, the objectives of the study were explained to study participants, and clarification was given before starting socio-demographic data and sample collection. To keep confidentiality of information from participants, no personal identifiers were used and the collected data were not available to anyone except for the investigator and the study was conducted according to the Declaration of Helsinki.

## Results

### Socio-demographic characteristics of study participants

In this study, a total of 200 cancer patients had included at University of Gondar cancer treatment center during the study period. Out of these, 67.5% (135/200) were males. The mean age of the study participants was 22.25 years with a standard deviation of ± 22.55 with an age range of 2 months– 83 years and 30.5% (61/200) of the study participants belonged to 5–15 years of age. The majorities, 56% (113/200) cancer patients were pediatrics, and 26.5% (53/200) of them

belong under five years of age. In this study, most of the study participants had a history of fever and antibiotic therapy, 92% (185/200) and 28% (56/200), respectively. The socio-demographic characteristics of study participants are presented (Table 1).

## Phenotypic characteristics of the recovered isolates

A total of 200 blood samples were analyzed, and the prevalence of culture-positive bloodstream infection among cancer patients was 27% (54/200) (**Fig 1**), and six different types of bacterial isolates and yeast cells were isolated. Of these, *S. aureus* (31.5%, 17/54) was the frequently isolated bacteria followed by Coagulase-negative staphylococci (CoNS), (29.6%, 16/54), *K. pneumoniae*, (14.8%, 8/54), and *P. aeruginosa*, (9.3%, 5/54) (**Fig 2**). Regarding the phenotypic characteristics of *S. aureus*, isolates were identified based on their morphology and biochemical tests. Microscopically, the bacteria appeared as Gram-positive cocci, non-motile, and non-sporulated cocci. The bacteria grew well on mannitol salt agar and gave a golden yellow colony due to mannitol fermentation and β-hemolytic colonies on blood agar. Biochemically, all isolates were positive for catalase and coagulase tests. Concerning the distribution of *S. aureus* in the examined samples, the total prevalence of *S. aureus* was 31.5% (17/54).

The phenotypic characteristics of *K. pneumoniae* isolates were identified based on their morphology and biochemical characteristics. Microscopically, the bacteria looked like non-motile Gram-negative rods. The bacteria grew well on MacConkeys's agar and gave characteristics of large mucoid, pink colonies due to lactose fermentation. Biochemically, all isolates were positive for lactose fermentation, citrate-utilization, lysine decarboxylase test, and urease tests. Simultaneously, they were negative for the indole test and $H_2S$ production. The bacteriological inspection showed that the total prevalence of *K. pneumoniae* was 14.8% (8/54).

Moreover, the bacteriological examination of *P. aeruginosa* revealed that all isolates were motile, Gram-negative bacilli, arranged in double or short chains. The isolates were positive for catalase and citrate test while negative for $H_2S$ production, urease, and indole test. Based on the morphological and biochemical characteristics, all isolates identified as *P. aeruginosa*, and the bacterium grew on MacConkey's agar and showed fat, smooth, non-lactose

**Table 1. Socio-demographic characteristics of study participants from March-July 2021.**

| Independent variables | | Frequency (N) | Percent (%) |
|---|---|---|---|
| **Gender** | Male | 135 | 67.5 |
| | Female | 65 | 32.5 |
| **Age in year** | <5 | 53 | 26.5 |
| | 5–15 | 61 | 30.5 |
| | 15–30 | 35 | 17.5 |
| | 30–45 | 18 | 9 |
| | 45–60 | 13 | 6.5 |
| | >60 | 20 | 10 |
| **Patient category** | Pediatric | 113 | 56.5 |
| | Adult | 87 | 43.5 |
| | Neonate | 0 | 0 |
| **Fever** | Yes | 185 | 92 |
| | No | 15 | 8 |
| **Antibiotic therapy** | Yes | 56 | 28 |
| | No | 144 | 72 |
| **Sign of microbial growth** | Yes | 54 | 27 |
| | No | 146 | 73 |

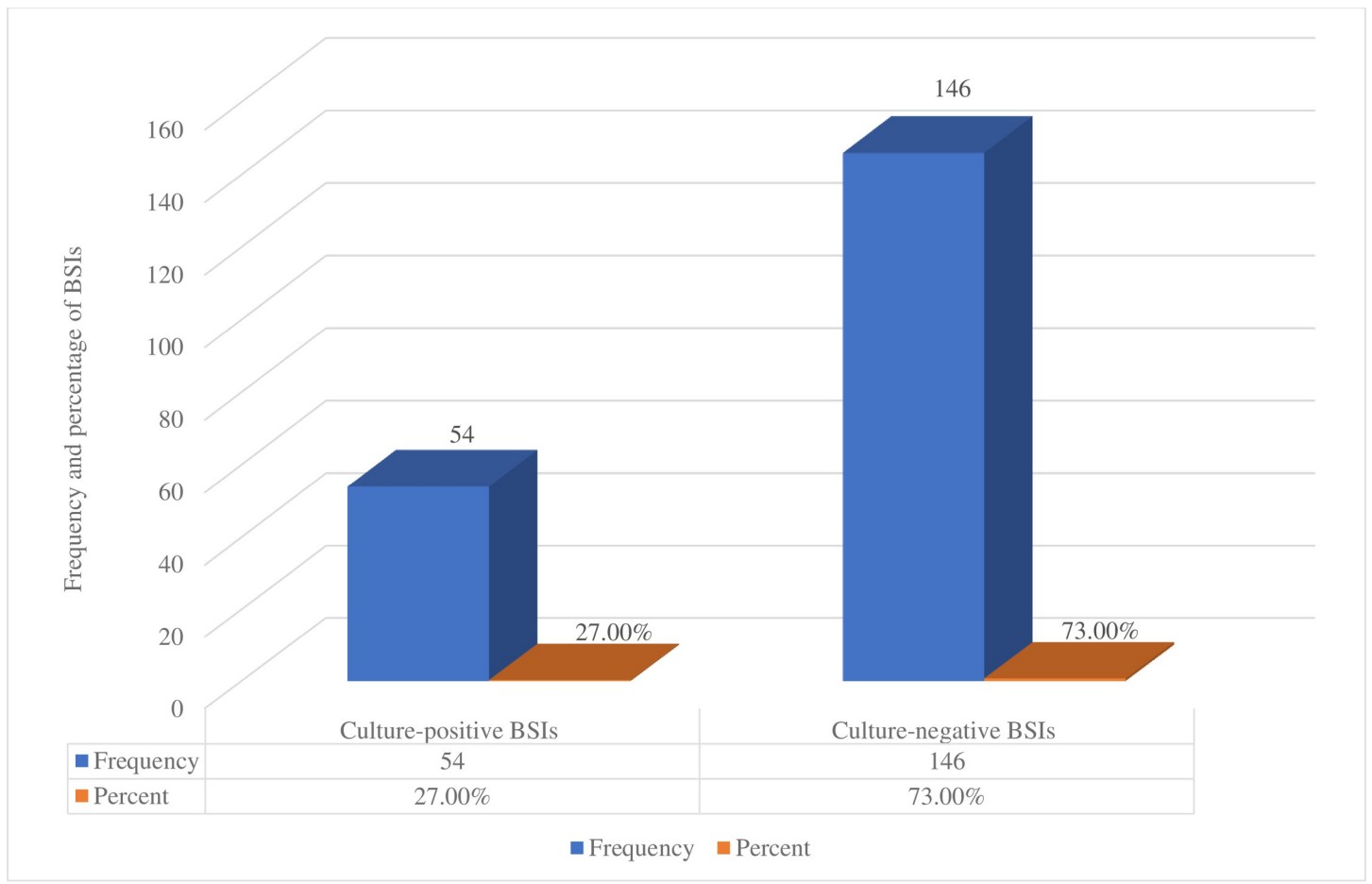

**Fig 1. Blood stream infections among cancer patients at University of Gondar comprehensive specialized hospital from March-July 2021.**

fermenting colonies. The isolates of *P. aeruginosa* displayed a bluish-green pigmented large colonies with characteristic "fruity" odor on culture media at 37˚C for 24 hours. The prevalence of *P. aeruginosa* was 9.3% (5/54) in the examined samples.

On the other hand, the proportion of the Gram-positive bacteria was 61.1% (33/54), and the predominant Gram-positive bacterium was *S. aureus* which was responsible for 51.5% (17/33) of BSIs, and fungal isolates were 7.4% (4/54). The proportion of the Gram-negative bacteria was 31.5% (17/54), and among the Gram-negative isolates, *K. pneumoniae* was the dominant Gram-negative bacteria 47.1% (8/17). The prevalence of bloodstream infection caused by Gram-positive bacteria (61.1%, 33) was higher than Gram-negative bacteria (31.5%, 17). Among patients who had BSI's, the highest prevalence of bloodstream infection was observed among males, 66.7% (36/54), and less than 15 years of study participants, 42.6% (23/54). Culture-positive bloodstream infection was high in pediatric cancer patients 88.9% (48/54) compared to adult cancer patients. The prevalence of culture-positive bloodstream infection among cancer patients who had a history of antibiotic therapy was 27.8% (15/54).

## Risk factors for bloodstream infection

Categorical variables were analyzed using Pearson chi-square. Based on this statistical method, age category and pediatric patients were significantly associated with bloodstream infection

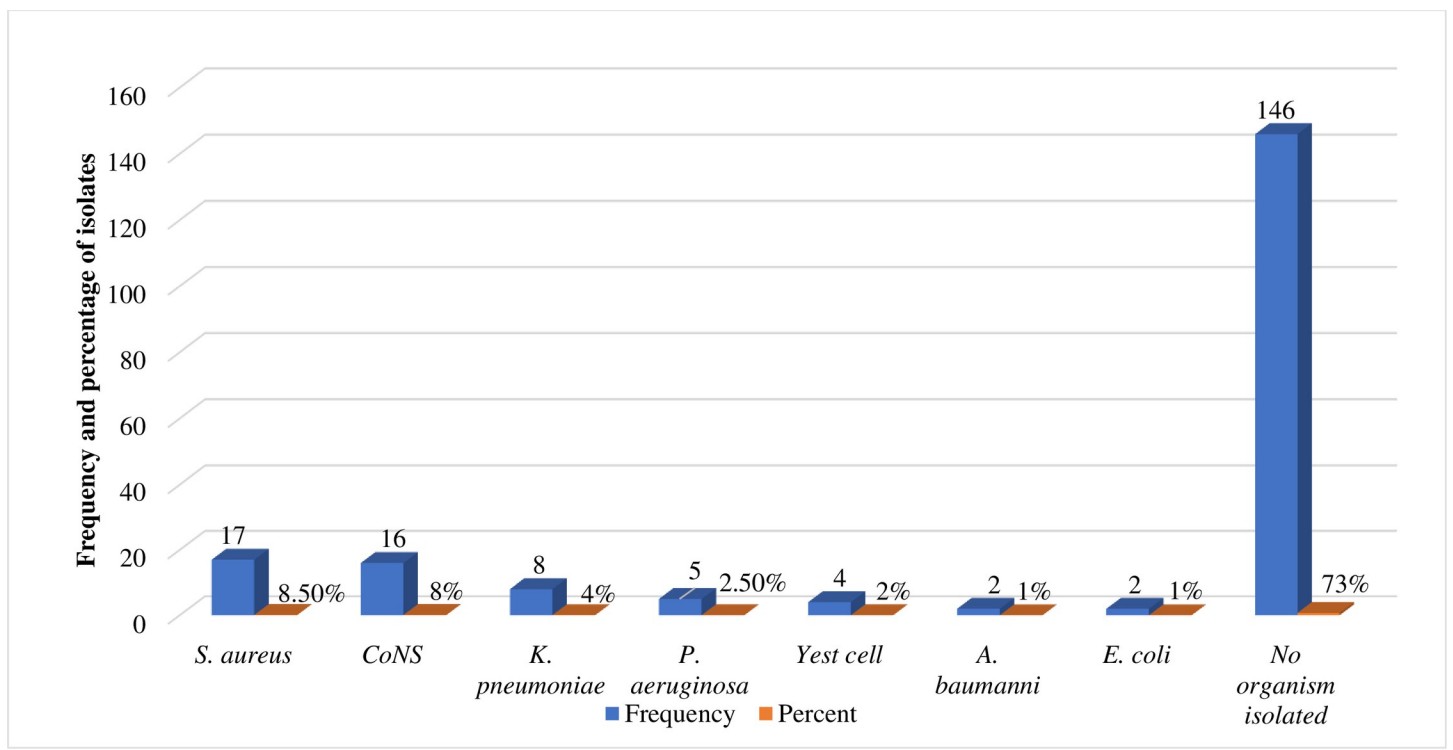

**Fig 2. Frequency of isolated organisms among cancer patients at University of Gondar comprehensive specialized hospital from March-July 2021.**

among cancer patients. Pediatric study participants were more venerable to bloodstream infection (P = 0.000) than adult participants. Moreover, Cancer patients less than 45 years of age were more at risk for bloodstream infection (P < 0.003) than adult cancer patients whose age was greater than 45 years. However, fever (P = 0.8) and antibiotic therapy (P = 0.8) were not statistically significant with bacterial bloodstream infections (Table 2).

## Antimicrobial susceptibility patterns of bacterial isolates

Bacterial antimicrobial susceptibility tests done for bacterial isolates, and meropenem (100%), amikacin (100%), piperacillin/tazobactam (72.3%), ceftazidime (73.5%) are effective antimicrobial agents to treat Gram-negative bacterial isolates while cefoxitin (81.2%), and penicillin (70.5%) effective against Gram-positive isolates. Moreover, most Gram-negative and Gram-positive bacterial isolates are sensitive to gentamycin (75.9%). All *K. pneumoniae* isolates were susceptible to meropenem (100%, 8), while the majority of *K. pneumoniae* isolates were sensitive to amikacin (75%, 6) and augmentin (75%, 6). The majority of *P. aeruginosa* isolates were susceptible to amikacin (100%, 5), meropenem (100%, 5), and ceftazidime (80%, 4). Additionally, the majority of the *S. aureus* isolates were susceptible to cefoxitin (81.2%) and the rest 18.8% of *S. aureus* isolates were MRSA. Furthermore, *S. aureus* isolates were susceptible to penicillin (70.5%), and gentamycin (70.5%) (Table 3).

## Multidrug resistance pattern of bacterial isolates

Multidrug-resistant isolates refer to an isolate resistant to at least one antibiotic in three or more antimicrobial classes. In this study, the prevalence of MDR isolates was 17.1% (n = 6). Multidrug resistance was detected among Gram-negative and Gram-positive bacteria, 83.3%, and

**Table 2. Associations of independent variables and BSI among cancer patients at UoGCSH from March-July 2021.**

| Independent variables | | Blood stream infection (BSI) | | | | | |
|---|---|---|---|---|---|---|---|
| | | Yes | | No | | | |
| | | Frequency | % | Frequency | % | person $x^2$ | p-value |
| Gender | Male | 36 | 26.7 | 99 | 73.3 | 0.05 | 0.8 |
| | female | 19 | 29.2 | 46 | 70.8 | 1 | |
| Age | <5 | 23 | 43.4 | 30 | 50.6 | 9.1 | **0.003** |
| | Other | 32 | 21.8 | 115 | 78.2 | 1 | |
| | 5–15 | 24 | 39.3 | 37 | 60.7 | 6.15 | **0.012** |
| | Other | 31 | 22.3 | 108 | 77.7 | 1 | |
| | 15–30 | 2 | 5.7 | 33 | 94 | 10.1 | **0.001** |
| | other | 53 | 32 | 112 | 67.9 | 1 | |
| | 30–45 | 23 | 11.1 | 43.4 | 56.6 | 9.1 | **0.003** |
| | other | 32 | 21.8 | 115 | 78.2 | 1 | |
| | 45–60 | 0 | 0 | 0 | 0 | - | - |
| | >60 | 4 | 20 | 16 | 80 | 0.6 | 0.4 |
| | Other | 51 | 28.3 | 129 | 71.7 | 1 | |
| Patient category | pediatric | 50 | 44.2 | 63 | 53.8 | 34.64 | **0.000** |
| | adult | 5 | 5.7 | 82 | 94.3 | 1 | |
| Fever | Yes | 54 | 29.2 | 131 | 70.8 | 3.5 | 0.06 |
| | No | 1 | 6.7 | 14 | 93.3 | 1 | |
| Antibiotic therapy | No | 40 | 27 | 108 | 73 | 0.064 | 0.8 |
| | Yes | 15 | 28.8 | 37 | 71.2 | 1 | |

16.7%, respectively. Gram-negative bacterial isolates showed a high prevalence of MDR than Gram-positive isolates. Out of 50 bacterial isolates, 37.5% (n = 3) of the *K. pneumoniae* isolates were found multidrug-resistant. MDR was observed in *P. aeruginosa* (20%, 1), *A. baumannii* (50%, 1), and *S. aureus* (6.25%, 1) (Table 4). Moreover, *K. pneumoniae* showed extensively drug-resistant (XDR) that is non-susceptibility to at least one agent in tetracycline's, aminoglycosides, cephalosporins, fluoroquinolones, folate pathway inhibitors, and penicillin's (Table 5).

## Discussion

Bloodstream infections are a cause of significant morbidity and mortality in cancer patients. To our knowledge, this study is the first comprehensively to address microbial spectrum and drug-susceptibility patterns of pathogens causing BSIs in cancer patients in the study area. The prevalence of bloodstream infection in this study was 27%, which is higher than a study conducted in Iraq 21.4% [4], Turkey 14.5% [6], Ethiopia 13.2% [12], Ghana 22% [49], Iran 7.95% [50], and South Africa 13% [51]. However, lower than a study reported in India 33.3% [52], Qatar 38.7% [16], India 56% [53], Zimbabwe 35.2% [54], Pakistan 43.6% [55], Japan 71.9% [56], Turkey 50.5% [57], and Ethiopia 71% [58]. Since the etiology of bacterial infections differs between centers and countries and changes over time, the difference in epidemiology of bacterial infections might be due to differences in geographic location, sample size, diagnostic technique, and infection control policies in the hospital.

Gram-positive bacteria were the predominant which was 66.1%. This finding was higher than a study conducted in Iraq 24.9% [4], Taiwan 42.3% [5], Korea 32.7% [12], India 31.8% [13], Greece 44% [14], Qatar 55% [16], Pakistan 43% [17], Turkey 10.9% [18], Zimbabwe 56% [54], Pakistan 15% [55], and Japan 15.3% [56] while Gram-negative bacteria were 31.4%, which is lower than a study conducted in Korea 55.6% [12], India 68% [13], Iraq 75% [4],

**Table 3. Antimicrobial susceptibility patterns of bacterial isolates among cancer patients at UoGCSH from March-July 2021.**

| Antibiotic patterns | | Types of isolated organism | | | | |
|---|---|---|---|---|---|---|
| | | K. pneumoniae | E. coli | P. aeruginosa | A. baumannii | S. aureus |
| | | No. (%) | No. (%) | No. (%) | No. (%) | No. (%) |
| TET | S | 6 (75) | 1 (50) | - | - | - |
| | R | 2 (25) | 1 (50) | - | - | - |
| TOB | S | 5 (62.5) | 2 (100) | 3 (60) | 2 (100) | - |
| | R | 3 (37.5) | - | 2 (40) | - | - |
| AMIK | S | 6 (75) | 2 (100) | 5 (100) | 1 (50) | - |
| | R | 2 (25) | - | - | 1 (50) | - |
| GEN | S | 4 (50) | 1 (50) | 5 (100) | 2 (100) | 12 (70.5) |
| | R | 4 (50) | 1 (50) | - | - | 4 (23.2) |
| TRM/SXT | S | 5 (62.5) | 2 (100) | - | - | - |
| | R | 3 (37.5) | - | - | - | - |
| MER | S | 8 (100) | 2 (100) | 5 (100) | - | - |
| | R | - | - | - | 2 (100) | - |
| CIP | S | 4 (50) | 2 (100) | 3 (60) | 1 (50) | - |
| | R | 4 (50) | - | 2 (40) | 1 (50) | - |
| FOX | S | - | - | - | - | 13 (81.25) |
| | R | - | - | - | - | 2 (12.5) |
| CRO | S | 4 (50) | 2 (100) | - | 1 (50) | - |
| | R | 4 (50) | - | - | 1 (50) | - |
| CAZ | S | 5 (62.5) | 2 (100) | 4 (80) | 1 (50) | - |
| | R | 3 (37.5) | - | 2 (40) | 1 (50) | - |
| P/TZ | S | 6 (75) | 1(50) | 4 (80) | 1 (150) | - |
| | R | 2 (25) | 1 (50) | 1 (20) | 1 (50) | - |
| AMC/CLA | S | 6 (75) | 1 (50) | - | - | - |
| | R | 2 (25) | 1 (50) | - | - | - |
| AMP | S | - | 1 (50) | - | - | - |
| | R | - | 1 (50) | - | - | - |
| PEN | S | - | - | - | - | 12 (70.5) |
| | R | - | - | - | - | 4 (23.4) |
| MDR | Y | 3 (37.5) | - | 1 (20) | 1 (50) | 1 (6.25) |
| | N | 5 (62.5) | 2 (100) | 4 (80) | 1 (50) | 15 (93.75) |

TET-Tetracycline; TOB-Tobramycine; AMIK-Amikacin; GEN-Gentamycin; TRM/SXT-Trimethoprim-sulfamethoxazole; MER-Meropenem; CIP-Ciprofloxacillin; FOX- Cefoxitin; CRO-Ceftriaxone; CAZ-Ceftazidime; P/TZ-Piperacillin/tazobactam; AMC/CLA-Amoxicillin/clavulanate; AMP-Ampicillin; PEN-Penicillin; MDR-Multidrug resistance.

Taiwan 44% [5], Greece 51.5% [14], Turkey 52.6% [18], Ghana 43%% [49], Zimbabwe 42% [54], and Pakistan 85% [55]. But Gram-negative bacteria in this study were higher than a study conducted in Turkey 13.2% [56]. Many studies reveal that Gram-negative bacteria were highly responsible for BSI among cancer patients. But Gram-positive cocci were predominant for bloodstream infections. Our result is supported by a study conducted in India [13], Sweden [59], South Africa [51], Addis Abeba, Ethiopia [58]. The possible reason for the preponderance of Gram-positive bacteria in this study might be the increased use of indwelling catheters. Moreover, chemotherapy-induced mucositis and the use of both prophylactic and empiric antibiotic regimens targeting Gram-negative bacteria diminishes recovery of Gram-negative pathogens, while selecting for Gram-positive bacteria were reported.

**Table 4. Frequency and multidrug resistance pattern of bacterial isolates at UoGCSH from March-July 2021.**

| Isolated organism | | Frequency (N) | Percent (%) |
|---|---|---|---|
| Gram stain result | Gram-positive cocci | 33 | 16.5 |
| | Gram-negative rod | 17 | 8.5 |
| | Yeast cell | 4 | 2 |
| | No organism | 146 | 73 |
| Types of isolated organism | *CONS* | 17 | 8.5 |
| | *S. aureus* | 16 | 8 |
| | *K. pneumoniae* | 8 | 4 |
| | *P. aeruginosa* | 5 | 2.5 |
| | *Acinetobacter* species | 2 | 1 |
| | *E. coli* | 2 | 1 |
| | Yeast cells | 4 | 2 |
| Multidrug resistance | *K. pneumoniae* | 3 | 50 |
| | *P. aeruginosa* | 1 | 16.6 |
| | *Acinetobacter* species | 1 | 16.6 |
| | *S. aureus* | 1 | 16.6 |

The bacterial infections epidemiology among cancer patients has changed significantly with the passage of time in recent decades and has shifted from Gram-negative to Gram-positive pathogens [3]. Bacterial infections, particularly those due to Gram-positive bacteria, continue to predominate in patients with cancer [60, 61]. The most probable explanation for this shift is the use of prophylactic antibiotics, the extensive use of central venous catheters, and the intensity and form of cancer treatment [62]. On the other hand, similar reports from developing countries still reveal the Gram-negative predominance [47, 63]. The relatively lower use of indwelling medical devices, as well as low utilization of prophylactic antibiotic regimens in cancer patients, have been indicated as factors in the dominance of Gram-negative bacteria in developing countries [64]. Although these conditions hold true in our setting, the predominance of Gram-positive bacteria in our case might be due to Gram-positive bacteria have a strong adhesion and biofilm forming abilities and they frequently isolated in the hospital area.

**Table 5. MDR, XDR and PDR pattern of bacterial isolates at UoGCSH from March-July 2021.**

| Bacterial isolates | MDR (%) | R | XDR (%) | PDR (%) | Antimicrobial Classes |
|---|---|---|---|---|---|
| *S. aureus* | 1/17 (5.9) | GEN, FOX, PEN | - | - | Aminoglycosides, Cephamycins, Penicillin's |
| *K. pneumoniae* | 3/8 (37.5) | TET, TOB, AMIK, GEN, TRM/SXT,CIP, CRO, CAZ, P/TZ, AMC/CLA | 3 (37.5) | - | Tetracycline's, Aminoglycosides, Cephalosporins, Fluoroquinolones, Folate pathway inhibitors, Penicillin's |
| *P. aeruginosa* | 1/5 (20) | TOB, CIP, CAZ, P/TZ | - | - | Aminoglycosides, Fluoroquinolones, Cephalosporins, Penicillin's |
| *A. baumannii* | 1/2 (50) | AMIK, CIP, CRO, CAZ, P/TZ, | - | - | Aminoglycosides, Fluoroquinolones, Cephalosporins, Penicillin's |
| *E. coli* | 0/2 (0) | TET, GEN, P/TZ, AMC/CLA, AMP | - | - | Tetracycline's, Aminoglycosides, Penicillin's |

TET-Tetracycline; TOB-Tobramycine; AMIK-Amikacin; GEN-Gentamycin; TRM/SXT-Trimethoprim-sulfamethoxazole; MER-Meropenem; CIP-Ciprofloxacillin; FOX- Cefoxitin; CRO-Ceftriaxone; CAZ-Ceftazidime; P/TZ-Piperacillin/tazobactam; AMC/CLA-Amoxicillin/clavulanate; AMP-Ampicillin; PEN-Penicillin;

MDR-Multidrug resistance, XDR—extensively drug-resistant, PDR-pandrug-resistant.

Note

R—Resistant to tested antibiotics.

MDR—The isolate is non-susceptible to at least 1 agent in $\geq$ 3 antimicrobial categories.

XDR—The isolate is non-susceptible to at least 1 agent in all but 2 or fewer antimicrobial categories.

PDR—Non-susceptibility to all agents in all antimicrobial categories.

Among Gram-positive bacteria, *S. aureus* (51.5%) was the commonest isolate, which is in line with a study conducted in Addis Abeba [58], Sudan [65], North America [66], and Japan [67]. However, this finding was higher in a study conducted in South Korea 8.9% [12], India 12.6% [13], Sweden 4% [15], Zimbabwe 8% [54], and Turkey 2% [57]. In our study, *S. aureus* was 81% susceptible to cefoxitin and 70.5% for gentamycin [68], which was in line with a study conducted in India [13] but many scholars agreed that these antibiotics are highly resistant which indicates MRSA [17, 54, 69]. *S. aureus* was the major microbial driver of sepsis amongst cancer patients and the treatment of *S. aureus* infections is complicated due to the bacteria being highly adaptable to resist many antibiotics, and no vaccine is available [70]. Gram-negative bacteria were 100% susceptible to amikacin and meropenem and 73% susceptible to ceftazidime. This finding was similar to a study conducted in Taiwan 100% to meropenem [5], in Qatar 100% susceptible to amikacin and carbapenem [16], in Zimbabwe which was 100% susceptible to amikacin and meropenem [54]. Infections caused by *S. aureus*, *K. pneumoniae*, and *P. aeruginosa* are difficult to treat because they can acquire resistance to many antibiotics by modification of antibiotic molecules, decreased antibiotic penetration or increased efflux, and bypassing of target site [22, 71, 72].

The overall prevalence of multidrug resistance was 17.1% and lower than a study conducted in Taiwan 44% [5], Qatar 28.4% [16], Ghana 44.9% [49], Mexico 39.2% [73], Australia 50% [74], and Italy 25.7% [75]. This variation might be due to the difference in empirical treatment and prophylaxis in the study area. Besides this, Gram-negative bacteria showed a high MDR pattern, 83% and about 40% of Gram-negative bacteria in the current study were multidrug-resistant. This finding is comparable with a study conducted in Pakistan [17]. This high rate of multidrug resistance is an important public health concern that needs to be addressed. Studies have shown that infections caused by MDR bacteria have higher morbidity and mortality rates [66, 76]. Therefore, appropriate use of antibiotics should be ensured in our setting. Treatment of the patient should be based on the antimicrobial susceptibly pattern of the etiologic agent. Our finding was not agreed with a study conducted in Qatar which showed Gram-positive cocci had a high rate of MDR pattern, 32% [16]. Being pediatrics patients were statically associated with bloodstream infection (p = 0.000). The reason for this could be pediatric cancer patients will have a low immune system and poor personal hygienic practice.

## Limitations of the study

As a limitation of this study, selection bias might be introduced because of the convenient sampling technique. Moreover, because of budget constraints and lack of facility, we were unable to perform molecular analysis such as detection of antibiotic resistance genes and virulence genes using PCR, and gene sequencing to associate between phenotypic and genotypic multidrug resistance in the recovered isolates.

## Conclusion and recommendations

BSI's remains an important health problem in cancer patients and Gram-positive bacteria were more common as etiologic agents of BSIs in cancer patients. *S. aureus* was the dominant bacteria followed by CoNS, *K. pneumoniae*, and *P. aeruginosa*. Pediatric cancer patients are at high risk for bloodstream bacterial infections and significant MDR bacterial isolates were detected. Meropenem and cefoxitin were the most effective antibiotics for the treatment of Gram-negative and Gram-positive bacteria, respectively. Routine bacterial surveillance and study of their resistance patterns may guide successful antimicrobial therapy and improve the quality of care. Therefore, strict regulation of antibiotic stewardship and infection control programs should be considered in the study area.

## Supporting information

**S1 Table. The AST interpretation chart (extracted from CLSI, 2021).**
(DOCX)

**S2 Table. Antimicrobial susceptibility pattern collection form.**
(DOCX)

**S1 Dataset. Dataset used for analysis of the result.**
(SAV)

## Acknowledgments

We would like to thank all participants of this research, University of Gondar Hospital, the staff members of Medical Microbiology, School of Biomedical and Laboratory Sciences for their contribution to the maturation and the success of this research.

## Author Contributions

**Conceptualization:** Minichil Worku, Gizeaddis Belay.

**Data curation:** Minichil Worku, Gizeaddis Belay, Abiye Tigabu.

**Formal analysis:** Minichil Worku, Gizeaddis Belay.

**Funding acquisition:** Gizeaddis Belay.

**Investigation:** Minichil Worku, Gizeaddis Belay.

**Methodology:** Minichil Worku, Gizeaddis Belay, Abiye Tigabu.

**Project administration:** Gizeaddis Belay.

**Resources:** Gizeaddis Belay.

**Software:** Gizeaddis Belay.

**Supervision:** Minichil Worku.

**Validation:** Minichil Worku, Abiye Tigabu.

**Visualization:** Minichil Worku, Abiye Tigabu.

**Writing – original draft:** Minichil Worku, Gizeaddis Belay.

**Writing – review & editing:** Abiye Tigabu.

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
