## [Decision Letter · Decision Letter 0]

10 Jan 2022

PONE-D-21-38142Bacterial bloodstream infections and antimicrobial susceptibility patterns among cancer patients at University of Gondar comprehensive specialized hospital.PLOS ONE

Dear Dr. Tigabu,

Thank you for submitting your manuscript to PLOS ONE. After careful consideration, we feel that it has merit but does not fully meet PLOS ONE’s publication criteria as it currently stands. Therefore, we invite you to submit a revised version of the manuscript that addresses the points raised during the review process.

ACADEMIC EDITOR: Please revise the manuscript according to the reviewer comments. A major revision is required./>==============================

We look forward to receiving your revised manuscript.

Kind regards,

Abdelazeem Mohamed Algammal, Prof, Ph.D

Academic Editor

PLOS ONE

Journal Requirements:

4.  We noticed you have some minor occurrence of overlapping text with the following previous publication(s), which needs to be addressed:

- https://bmcresnotes.biomedcentral.com/track/pdf/10.1186/s13104-019-4520-9.pdf

- https://link.springer.com/article/10.1007%2Fs15010-004-3049-5

- https://www.dovepress.com/microbial-spectrum-and-drug-resistance-profile-of-isolates-causing-blo-peer-reviewed-fulltext-article-IDR 

The text that needs to be addressed involves the Discussion.

In your revision ensure you cite all your sources (including your own works), and quote or rephrase any duplicated text outside the methods section. Further consideration is dependent on these concerns being addressed.

Reviewers' comments:

Reviewer's Responses to Questions

**Comments to the Author**

1. Is the manuscript technically sound, and do the data support the conclusions?

Reviewer #1: Partly

Reviewer #2: Yes

2. Has the statistical analysis been performed appropriately and rigorously? 

Reviewer #1: Yes

Reviewer #2: Yes

3. Have the authors made all data underlying the findings in their manuscript fully available?

Reviewer #1: Yes

Reviewer #2: Yes

4. Is the manuscript presented in an intelligible fashion and written in standard English?

Reviewer #1: No

Reviewer #2: No

5. Review Comments to the Author

Reviewer #1: Comments to authors:

-The current study is interesting; however, the authors should address the following comments to improve the quality of the manuscript:

Title:

I think the work would benefit from the title that contains the main conclusion of the study (should be derived from the conclusion). Please modify the title.

Abstract:

- The abstract must illustrate the used methods and the most prevalent results (give more hints about methods and results). Besides, rephrase the aim of the work and the main conclusion of your findings.

Introduction: (it needs to be more informative)

-Give a hint about the virulence factors, different infections caused by S. aureus, K. pneumoniae, and Pseudomonas aeruginosa, and the mechanism of disease occurrence.

- The authors should illustrate the public health importance concerning the emergence of multidrug-resistant (MDR) bacterial pathogens that reflect the necessity of new potent and safe antimicrobial agents. Several studies proved the widespread MDR- bacterial pathogens;

Authors could add the following paragraph:

Multidrug resistance has been increased all over the world that is considered a public health threat. Several recent investigations reported the emergence of multidrug-resistant bacterial pathogens from different origins including humans, birds, cattle, and fish that increase the need for routine application of the antimicrobial susceptibility testing to detect the antibiotic of choice as well as the screening of the emerging MDR strains. You should cite the following valuable studies:

1.PMID: 33177849

2. PMID: 32397408

3.PMID: 32994450

4. PMID: 32497922

5.PMID: 33061472

6.PMID: 33947875

7.PMID: 34445951

8.PMID: 33188216

9.https://doi.org/10.1016/j.aquaculture.2021.737643

10.PMID: 30150182

-Rephrase the aim of the work to be clear and better sound.

Material and methods: Illustrate your methods with subtitles:

-Add this subtitle: Bacterial Isolation and identification:

•Discuss in detail the methods of isolation and identification of S.aureus and MRSA. Besides, specific references should be added.

•Add the company, city, and country of the used bacterial media and reagents that were used in the biochemical identification of isolates. Also, enumerate all used biochemical reactions.

- Antimicrobial susceptibility testing:

•Illustrate the antimicrobial classes of the tested antimicrobial agents within the text.

•The authors are advised to classify the tested isolates to MDR , XDR, and PDR as described by Magiorakos et al.

Magiorakos AP, Srinivasan A, Carey RB, Carmeli Y, Falagas ME, Giske CG, et al. Multidrug-resistant, extensively drug-resistant and pandrug-resistant bacteria: An international expert proposal for interim standard definitions for acquired resistance. Clin Microbiol Infect. 2012; 18:268–81. doi:10.1111/j.1469-0691.2011.03570.x.

- Why did you ignore the detection of antibiotic resistance genes in the recovered isolates??

•Please use PCR to detect antibiotic resistance genes and virulence genes, followed by gene sequencing if possible. Afterward, the correlation between phenotypic and genotypic multidrug resistance should be performed.

-Add more details about the software used in the statistical analyses.

-Results:

-Add this subtitle: Phenotypic characteristics of the recovered isolates.

•Illustrate in detail the phenotypic characteristics of the recovered isolates, especially, S. aureus, K. pneumoniae, and Pseudomonas aeruginosa.

-Antimicrobial susceptibility testing:

•-Illustrate in a new table the occurrence of MDR (Multidrug resistance) among the recovered isolates (illustrate the names of the antimicrobial classes and different antibiotics):

No. of strains%Type of resistance

R, MDR, and XDRPhenotypic multidrug resistance

(Antimicrobial classes and different antibiotics).The antibiotic -resistance genes

- Increase the resolution of all figures (it should be 600 dpi).

-Discussion:

- The authors are advised to illustrate the real impact of their findings without repetition of results.

-Illustrate the different mechanisms of antimicrobial resistance in S.aureus, K. pneumoniae, and Pseudomonas aeruginosa

-Conclusion

- Should be rephrased to be sounded. A real conclusion should focus on the question or claim you articulated in your study, which resolution has been the main objective of your paper?

Reviewer #2: Comments to authors:

- The current study has a significant impact, but it needs a major revision:

- The manuscript should be revised for grammar mistakes.

- Please write the scientific names of bacterial pathogens and genes in the correct form all over the manuscript and in the References section (should be italic).

-The title is broad, please modify the title.

- Add more details about the used methods and most prevalent results in the abstract.

-In the introduction: discuss the public health importance of S. aureus, K. pneumoniae, and Pseudomonas aeruginosa, and their virulence determinants.

-Improve the aim of work.

Methods:

-Explain the methods of isolation and identification in detail??

-Specific references should be added to all the used methods and techniques.

- Antimicrobial susceptibility testing: Add the manufacturing company, city, and country for the used reagents and antimicrobial discs.

-PCR based detection of virulence genes and antimicrobial resistance genes in the most prevalent retrieved bacterial species should be carried out if applicable (or addresses this point in the study limitations)

--Results:

- Discuss in detail the phenotypic characters of the recovered isolates: S. aureus, K. pneumoniae, and Pseudomonas aeruginosa.

-increase the resolution of different Figures: Please improve.

-PCR based detection of virulence genes and antimicrobial resistance genes in the most prevalent retrieved bacterial species should be carried out if applicable (or addresses this point in the study limitations)

-The correlation between the phenotypic and genotypic MDR should be performed.

-Discussion:

- Please improve.

-Please improve the main conclusion of the manuscript.

6. PLOS authors have the option to publish the peer review history of their article (what does this mean?). If published, this will include your full peer review and any attached files.

Reviewer #1: No

Reviewer #2: No

---

## [Author Response · Author response to Decision Letter 0]

18 Mar 2022

To: The Editor of PLOS ONE

Title: Bacterial profile and antimicrobial susceptibility patterns in cancer patients

Submission ID: PONE-D-21-38142

Corresponding author: Abiye Tigabu

Thank you for considering our article for revision and we are grateful to the editor and the reviewers for their constructive comments.

 Point by point response to editor comments

- We revised the manuscript according to PLOS ONE's style requirements.

- A statement that states about “consent from parents or guardians of the minors” added in the revised manuscript.

- We declared that all data underlying the findings described fully available, without accession numbers or DOIs necessary to access our data upon publication of the work, and the changes described in the cover letter.

4. We noticed you have some minor occurrence of overlapping text with the following previous publication(s), which needs to be addressed:

- https://bmcresnotes.biomedcentral.com/track/pdf/10.1186/s13104-019-4520-9.pdf

- https://link.springer.com/article/10.1007%2Fs15010-004-3049-5

- https://www.dovepress.com/microbial-spectrum-and-drug-resistance-profile-of-isolates-causing-blo-peer-reviewed-fulltext-article-IDR

The text that needs to be addressed involves the Discussion.

In your revision ensure you cite all your sources (including your own works), and quote or rephrase any duplicated text outside the methods section.

- Overlapping texts reviewed and corrected throughout the revised manuscript and the sources of information are cited. 

 Point by point response to Reviewer 1

 Title

1. I think the work would benefit from the title that contains the main conclusion of the study (should be derived from the conclusion). Please modify the title.

- The title modified in the revised manuscript accordingly.

Abstract

2. The abstract must illustrate the used methods and the most prevalent results (give more hints about methods and results). Besides, rephrase the aim of the work and the main conclusion of your findings.

- The methods used and the most prevalent findings of the study added in the revised manuscript. Additionally, the aim and the conclusion of the work rephrased.

Introduction

3. It needs to be more informative?

- Information from several studies added and the introduction rewritten to make informative in the revised manuscript.

4. Give a hint about the virulence factors, different infections caused by S. aureus, K. pneumoniae, and Pseudomonas aeruginosa, and the mechanism of disease occurrence.

- Virulence factors, different infections and disease mechanism caused by S. aureus, K. pneumoniae, and P. aeruginosa added in the revised manuscript.

5. The authors should illustrate the public health importance concerning the emergence of multidrug-resistant (MDR) bacterial pathogens that reflect the necessity of new potent and safe antimicrobial agents. Several studies proved the widespread MDR- bacterial pathogens;

Authors could add the following paragraph:

Multidrug resistance has been increased all over the world that is considered a public health threat. Several recent investigations reported the emergence of multidrug-resistant bacterial pathogens from different origins including humans, birds, cattle, and fish that increase the need for routine application of the antimicrobial susceptibility testing to detect the antibiotic of choice as well as the screening of the emerging MDR strains. You should cite the following valuable studies:

1. PMID: 33177849

2. PMID: 32397408

3. PMID: 32994450

4. PMID: 32497922

5. PMID: 33061472

6. PMID: 33947875

7. PMID: 34445951

8. PMID: 33188216

9. https://doi.org/10.1016/j.aquaculture.2021.737643

10. PMID: 30150182

- The paragraph recommended by the reviewer and the public health importance of MDR bacterial pathogens illustrated in the revised manuscript.

Material and methods

6. Illustrate your methods with subtitles.

- Methods stated as subtitles in the revised manuscript.

7. Add this subtitle: Bacterial isolation and identification

- The subtitle bacterial isolation and identification added in the revised manuscript. 

8. Discuss in detail the methods of isolation and identification of S. aureus and MRSA. Besides, specific references should be added.

- The detail methods of isolation and identification of S. aureus and MRSA revised accordingly. 

9. Add the company, city, and country of the used bacterial media and reagents that were used in the biochemical identification of isolates. Also, enumerate all used biochemical reactions.

- The company, city, and country name of the used culture media and reagents stated in the revised manuscript.

- Biochemical tests used in study enumerated and corrected in the revised manuscript.

Antimicrobial susceptibility testing

10. Illustrate the antimicrobial classes of the tested antimicrobial agents within the text.

- Antimicrobial classes of the tested antimicrobial illustrated in the revised manuscript.

11. The authors are advised to classify the tested isolates to MDR, XDR, and PDR as described by Magiorakos et al.

Magiorakos AP, Srinivasan A, Carey RB, Carmeli Y, Falagas ME, Giske CG, et al. Multidrug-resistant, extensively drug-resistant and pandrug-resistant bacteria: An international expert proposal for interim standard definitions for acquired resistance. Clin Microbiol Infect. 2012; 18:268–81. doi:10.1111/j.1469-0691.2011.03570.x.

- Tested isolates classified to MDR, XDR, and PDR as described by Magiorakos et al.

12. Why did you ignore the detection of antibiotic resistance genes in the recovered isolates? Please use PCR to detect antibiotic resistance genes and virulence genes, followed by gene sequencing if possible. Afterward, the correlation between phenotypic and genotypic multidrug resistance should be performed.

- Due to budget constraints and lack of facility, we were unable to perform molecular analysis such as detection of antibiotic resistance and virulence genes using PCR, and gene sequencing to associate between phenotypic and genotypic multidrug resistance in the recovered isolates and it is included as limitation in the revised manuscript.

13. Add more details about the software used in the statistical analyses.

- Software’s used for statistical analyses added in the revised manuscript.

Results

14. Add this subtitle: Phenotypic characteristics of the recovered isolates. 

- The subtitle “Phenotypic characteristics of the recovered isolates” added in the revised manuscript.

15. Illustrate in detail the phenotypic characteristics of the recovered isolates, especially, S. aureus, K. pneumoniae, and P. aeruginosa.

- Phenotypic characteristics of S. aureus, K. pneumoniae, and P. aeruginosa included in the revised manuscript.

16. Illustrate in a new table the occurrence of MDR (Multidrug resistance) among the recovered isolates (illustrate the names of the antimicrobial classes and different antibiotics):

No. of strains % Type of resistance

R, MDR, and XDR Phenotypic multidrug resistance

(Antimicrobial classes and different antibiotics). The antibiotic -resistance genes.

- A new table added to illustrate multidrug resistance, No. of strains and % type of resistance in the revised manuscript.

17. Increase the resolution of all figures (it should be 600 dpi). 

- The resolution of all figures increased the revised manuscript. 

Discussion

18. The authors are advised to illustrate the real impact of their findings without repetition of results. 

- The real impacts of the findings discussed and the repetition of results removed.

19. Illustrate the different mechanisms of antimicrobial resistance in S. aureus, K. pneumoniae, and Pseudomonas aeruginosa.

- The antimicrobial resistance mechanisms of S. aureus, K. pneumoniae, and P. aeruginosa illustrated in the revised version of the manuscript.

Conclusion

20. Should be rephrased to be sounded. A real conclusion should focus on the question or claim you articulated in your study, which resolution has been the main objective of your paper? 

- The conclusion of the finding rephrased in the revised manuscript.

 Point by point response to Reviewer 2

General comments

1. The current study has a significant impact, but it needs a major revision.

- The manuscript revised in detail throughout the revised manuscript.

2. The manuscript should be revised for grammar mistakes.

- Grammar mistakes and spelling problems corrected in the revised manuscript.

3. Please write the scientific names of bacterial pathogens and genes in the correct form all over the manuscript and in the References section (should be italic).

- Names of bacterial pathogens written correctly throughout the revised manuscript.

4. The title is broad, please modify the title.

- The title modified in the revised manuscript accordingly.

5. Add more details about the used methods and most prevalent results in the abstract.

- Methods used and the most prevalent findings of the study added in the abstract section of the revised manuscript. 

Introduction

6. Discuss the public health importance of S. aureus, K. pneumoniae, and Pseudomonas aeruginosa, and their virulence determinants.

- Public health importance and virulence determinants of S. aureus, K. pneumoniae, and P. aeruginosa added in revised manuscript.

7. Improve the aim of work.

- The aim of the study improved in the revised manuscript.

Methods

8. Explain the methods of isolation and identification in detail??

- Methods of isolation and identification explained in a clear and detail manner.

9. Specific references should be added to all the used methods and techniques.

- References added for the specific methods and techniques used.

10. Antimicrobial susceptibility testing: Add the manufacturing company, city, and country for the used reagents and antimicrobial discs.

- The names of manufacturing company, city, and country of used reagents and antimicrobial discs included in the revised manuscript.

11. PCR based detection of virulence genes and antimicrobial resistance genes in the most prevalent retrieved bacterial species should be carried out if applicable (or addresses this point in the study limitations)

- Due to budget constraints and lack of facility, we were unable to perform molecular analysis such as detection of antibiotic resistance and virulence genes using PCR, and gene sequencing to associate between phenotypic and genotypic multidrug resistance in the recovered isolates and it is included as limitation in the revised manuscript.

Results

12. Discuss in detail the phenotypic characters of the recovered isolates: S. aureus, K. pneumoniae, and Pseudomonas aeruginosa.

- Phenotypic characters of S. aureus, K. pneumoniae, and P. aeruginosa presented in detail in result section of the revised manuscript.

13. Increase the resolution of different Figures: Please improve.

- The resolution of figures improved in the revised manuscript.

14. PCR based detection of virulence genes and antimicrobial resistance genes in the most prevalent retrieved bacterial species should be carried out if applicable (or addresses this point in the study limitations)

- Due to budget constraints and lack of facility, we were unable to perform molecular analysis such as detection of antibiotic resistance and virulence genes using PCR, and gene sequencing to associate between phenotypic and genotypic multidrug resistance in the recovered isolates and it is included as limitation in the revised manuscript.

15. The correlation between the phenotypic and genotypic MDR should be performed.

- We were unable to associate the correlation between the phenotypic and genotypic MDR due to resource constraints and it is mentioned as limitation in the manuscript

Discussion

16. Please improve.

- The discussion rewritten and improved in the revised manuscript.

17. Please improve the main conclusion of the manuscript.

- The conclusion of the finding rephrased and improved in the revised manuscript.

---

## [Editor Report · Decision Letter 1]

30 Mar 2022

Bacterial profile and antimicrobial susceptibility patterns in cancer patients

PONE-D-21-38142R1

Dear Dr. Tigabu,

We’re pleased to inform you that your manuscript has been judged scientifically suitable for publication and will be formally accepted for publication once it meets all outstanding technical requirements.

Kind regards,

Abdelazeem Mohamed Algammal, Prof, Ph.D

Academic Editor

PLOS ONE

Additional Editor Comments (optional):

The authors have carried out significant changes to the manuscript. They have addressed most of the suggested corrections and comments. Really, it's an interesting study that has a significant impact. Now, the manuscript could be accepted.
---

## [Editor Report · Acceptance letter]

7 Apr 2022

PONE-D-21-38142R1 

Bacterial profile and antimicrobial susceptibility patterns in cancer patients 

Dear Dr. Tigabu:

I'm pleased to inform you that your manuscript has been deemed suitable for publication in PLOS ONE. Congratulations! Your manuscript is now with our production department. 

Kind regards, 

on behalf of

Professor Abdelazeem Mohamed Algammal 

Academic Editor

PLOS ONE